# Deep learning-enabled breast cancer hormonal receptor status determination from base-level H&E stains

Nikhil Naik [1 ✉], Ali Madani[1,4], Andre Esteva[1,4], Nitish Shirish Keskar[1], Michael F. Press [2], Daniel Ruderman [3], David B. Agus[3] & Richard Socher[1]

For newly diagnosed breast cancer, estrogen receptor status (ERS) is a key molecular marker used for prognosis and treatment decisions. During clinical management, ERS is determined by pathologists from immunohistochemistry (IHC) staining of biopsied tissue for the targeted receptor, which highlights the presence of cellular surface antigens. This is an expensive, time-consuming process which introduces discordance in results due to variability in IHC preparation and pathologist subjectivity. In contrast, hematoxylin and eosin (H&E) staining—which highlights cellular morphology—is quick, less expensive, and less variable in preparation. Here we show that machine learning can determine molecular marker status, as assessed by hormone receptors, directly from cellular morphology. We develop a multiple instance learning-based deep neural network that determines ERS from H&E-stained whole slide images (WSI). Our algorithm—trained strictly with WSI-level annotations—is accurate on a varied, multi-country dataset of 3,474 patients, achieving an area under the curve (AUC) of 0.92 for sensitivity and specificity. Our approach has the potential to augment clinicians' capabilities in cancer prognosis and theragnosis by harnessing biological signals imperceptible to the human eye.

[1] Salesforce Research, 575 High St, Palo Alto, CA 94301, USA. [2] Department of Pathology, Keck School of Medicine, University of Southern California, 2011 Zonal Ave, Los Angeles, CA 90033, USA. [3] Lawrence J. Ellison Institute for Transformative Medicine, University of Southern California, 12414 Exposition Blvd, Los Angeles, CA 90064, USA. [4]These authors contributed equally: Ali Madani, Andre Esteva. ✉email: nnaik@salesforce.com

More than 2 million women across the world were diagnosed with breast cancer in 2018, resulting in 0.6 million deaths. A large majority of invasive breast cancers are hormone receptor-positive—the tumor cells grow in the presence of estrogen (ER) and/or progesterone (PR)[1–5]. Patients with hormone-receptor positive tumors often clinically benefit from receiving hormonal therapies, which target the ER signaling pathway. The US National Comprehensive Cancer Network guidelines mandate that hormone receptor status, including ER receptor status, be determined for every new breast cancer patient, as this is critical in clinical decision-making[6].

In the current diagnostic workflow, a patient's sample is thinly sectioned onto microscope slides for staining followed by visual diagnosis by a pathologist. Hematoxylin and eosin (H&E) staining is used for primary diagnosis, and specialized stains for molecular markers can be used for diagnostic confirmation and subtyping. For breast cancer, ERS is always assayed, as it is both a prognostic marker and predictive of endocrine therapy response. ERS is determined by visual inspection of slides stained using molecular immunohistochemistry (IHC) with an antibody targeting the ER receptor (Fig. 1a). This process has several limitations. IHC staining is expensive and time-consuming. The test output is expressed in terms of color: stain intensity, or percentage of cells that achieve a detectable stain intensity, or presence/absence of a stain. There can be significant variation in sample quality due to differences in tissue handling and fixation, antibody sources and clones, and technician skill levels[1,2]. Finally, the pathologists' decision-making process is inherently subjective and can result in human errors[3]. These factors lead to discordance in ERS determination; an estimated 20% of current IHC-based determinations of ER and PR testing may be inaccurate[3,4], placing these patients at risk for suboptimal treatment.

We find that the morphology of the tumor, captured in the H&E stain, contains predictive signal for the molecular marker status of the tumor and that a machine-learning (ML) algorithm can directly determine ERS from an H&E-stained whole slide image (Fig. 1a). The morphology is a reflection of the biology; in this case, the dependence on hormonal signaling and the arrangement of the cells one would predict may be different depending on the biology. We show that ML identifies histo-morphological feature groups within the tissue structure captured by an H&E stain that are predictive of molecular biomarkers (or biology) expressed in an IHC stain.

Our algorithm is trained with clinical ERS readily available from patient records and requires no additional pixel-level annotations. Recent studies[4,7,8] have shown promising performance for ERS determination from morphological stains, but are based on single-center datasets of tissue microarrays (TMAs). Creating TMAs is a manual process that requires pathologists to select regions of interest (ROIs) from the whole specimen[9,10]. In contrast, our method automatically selects ROIs from the total tissue field and demonstrates accurate results on a large, multi-country dataset of WSIs, which would make it feasible for pathologists to augment their standard clinical workflows with no additional manual steps. Other studies using TCGA and whole slide images have reported results on molecular biomarkers such as lung cancer[11].

Working with H&E has several advantages—it is significantly cheaper than IHC, it exhibits less variability across centers, and it is ubiquitously used in histopathology workflows globally. An automated ERS estimation method has the potential to reduce errors in breast cancer treatment and improve outcomes, and importantly reduce time to treatment decisions. Moreover, an algorithm that identifies discriminative morphological features for molecular markers has the potential to provide biological insights into how hormones drive tumor growth.

ML-driven histopathology methods have primarily relied on expensive, time-consuming, pixel-level pathologist annotations of

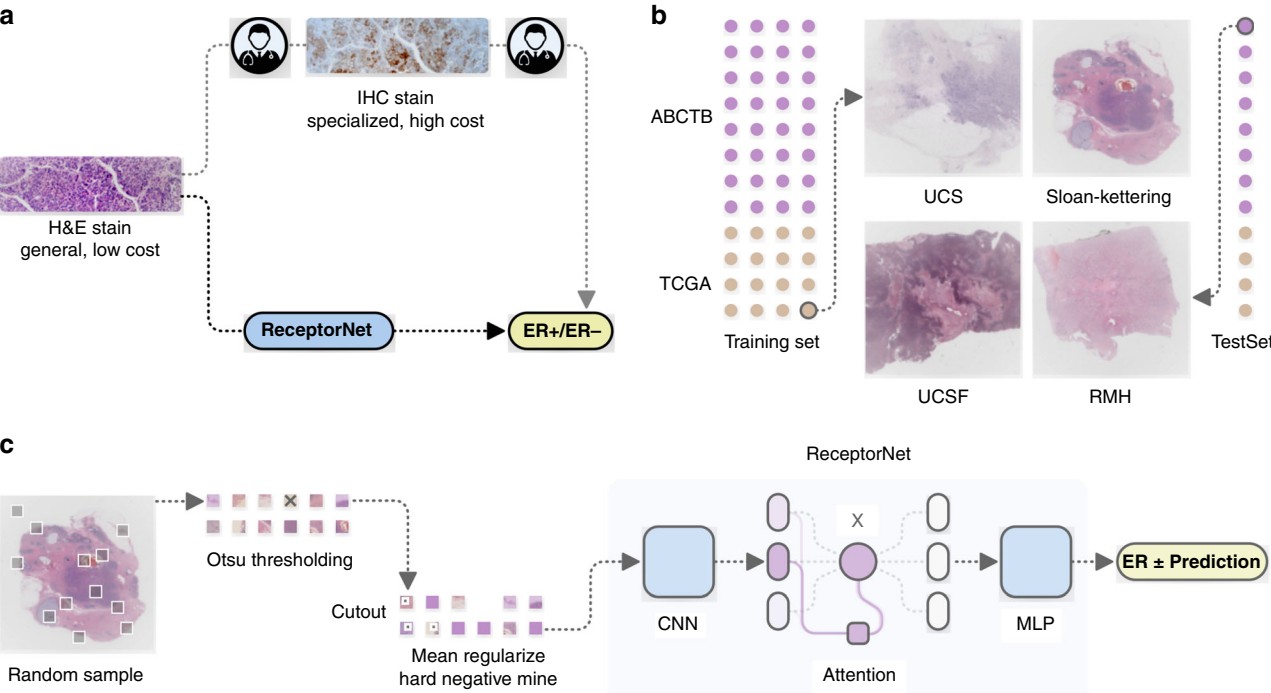

**Fig. 1 Estrogen receptor status (ERS) estimation from H&E stained whole slide images (WSI). a** In clinical practice, pathologists diagnose breast cancer from H&E stain, followed by estrogen receptor estimation from IHC stain. We show that a deep learning algorithm can accurately predict ERS directly from H&E stain. **b** Our algorithm, which we call ReceptorNet, is trained and evaluated on a diverse, multicenter dataset of 3474 patients with large variation in sample quality. **c** ReceptorNet is trained using patches directly sampled from WSI, with no pixel-level annotations. It automatically learns to pay attention to regions of the WSI important for ERS estimation.

whole slide images for training[11–15]. Since pathologists are not clinically trained to determine ERS from H&E images, they cannot provide manual annotations for this particular stain. However, if a tumor has been determined to be ER-negative (ER−) from an IHC stain, we can assume that their H&E WSI contains almost no discriminative features for ER-positivity. Conversely, if the patient has been determined to be ER-positive (ER+), we can assume that at least some regions of their H&E WSI contain discriminative features for ER-positivity. Thus, we can train models using H&E stains as input data, and IHC annotations as input data labels. This problem setup works well with multiple instance learning (MIL)[16]. MIL has been recently utilized for ML-driven histopathological diagnosis[17] and prognosis[18].

The goal of MIL is to learn from a training set consisting of labeled bags of unlabeled instances. A positively labeled bag contains at least one positive instance, while a negatively labeled bag contains only negative instances. A trained MIL algorithm is able to predict the positive/negative label for unseen bags. We utilize MIL to estimate ERS from a bag of tiles randomly selected from a WSI.

In addition to being accurate, an ML algorithm for ERS estimation should be interpretable—it should enable users to identify regions of the image that are important for decision-making. From a clinical perspective, interpretability is important for gaining physicians' trust, for building a robust decision-making system, and to help overcome regulatory concerns. From a scientific perspective, an interpretable model can help locate discriminative tiles in H&E images and identify histomorphological features that are correlated with hormone-driven cell growth. To achieve interpretability, we design an attention-based deep neural network that performs MIL[19], which we call ReceptorNet (Fig. 1c). ReceptorNet learns to assign high attention weights to tiles in the H&E image that have maximum discriminative capacity, and to assign low attention weights to tiles that are insignificant for this task. Analyzing attention weights assigned to different tiles allows us to determine which tiles were utilized to make ERS estimation.

ReceptorNet is trained to predict ER+ status from a bag of tiles randomly selected from a WSI at a resolution of 0.5 μm/pixel. The WSI is divided into $256 \times 256$ pixel-size nonoverlapping tiles. ReceptorNet consists of three interconnected neural networks which are trained together: (1) A feature extractor which converts each $256 \times 256$ pixel-size tile in a bag into a 512-dimensional feature vector, (2) an attention module that creates a 512-dimensional aggregate of feature vectors from all the tiles in the bag, attention-weighted based on discriminative power, and (3) a decision layer that computes the probability of this bag being positive from the aggregate feature vector. We improve on the existing attention-based MIL[19] algorithm by using cutout regularization[20], hard-negative mining[21], and a novel mean pixel regularization (see "Methods"). We train ReceptorNet to predict the probability of ER+ status through an iterative learning process. During testing, we sample multiple bags from a WSI and aggregate their probabilities to improve prediction accuracy.

## Results

**Quantitative evaluation.** To train and test ReceptorNet, we utilize two diverse datasets: the Australian Breast Cancer Tissue Bank (ABCTB) dataset containing 2535 H&E images from 2535 patients and The Cancer Genome Atlas (TCGA) dataset containing 1014 H&E images from 939 patients (Fig. 1b). TCGA images are obtained from 42 different tissue source sites from the USA, Poland, and Germany. Both datasets report the hormone receptor status determined by pathologists from IHC stains. Our

combined dataset has large variation in sample preparation, staining, and scanning quality (Supplementary Fig. 1). After removing images with excessive pen markings, we divide the combined dataset into a train set (2728 patients) and a test set (671 patients). After performing fivefold cross-validation on the train set, we train ReceptorNet using all slides from the train set and evaluate it on the test set slides. (See "Methods" for details on data preparation, training, and evaluation)

We report results using the area under the curve (AUC) for ER+/ER− binary classification, along with its 95% confidence interval (CI) computed using bootstrapping. ReceptorNet obtains an AUC of 0.899 (95% CI: 0.884–0.913) on the cross-validation of the train set and an AUC of 0.92 (95% CI: 0.892–0.946) on the test set (Fig. 2a). On the test set, our method obtains a positive predictive value (PPV) of 0.932 and an negative predictive value (NPV) of 0.741 at a threshold of 0.25. Since we do not have access to the original tissue material, it is not possible to perform an exact comparison between pathologists and our approach. We note that the results from the International Breast Cancer Study Group[3,4] dataset from a separate study, translate to a PPV of 0.92 and an NPV of 0.683 for concordance between primary institution and central testing on the combined dataset of premenopausal and postmenopausal patients. Validation against gold-standard data using central testing remains an important future direction.

To determine the importance of algorithm choice, we compared ReceptorNet with Meanpool and Maxpool, two traditional, widely used MIL algorithms[16]. Their performance is inferior to ReceptorNet on the test set ($p < 1 \times 10^{-4}$ for Meanpool and $p < 0.01$ for Maxpool, DeLong test). Meanpool obtains an AUC of 0.827 (95% CI: 0.786–0.866) and Maxpool obtains an AUC of 0.880 (95% CI: 0.846–0.912). To evaluate if ERS can be determined from individual patches alone, we created a model with the same basic architecture as ReceptorNet after removing the attention module and trained it on individual patches using binary cross-entropy. This model obtains an AUC of 0.760 (95% CI: 0.726–0.794), with evaluation averaged on 50 patches, indicating that aggregating information from multiple patches using an attention module leads to better performance. We also build a logistic regression model using pathologist-provided histological type and tumor grade, which are clinically determined from H&E. This model obtains an AUC of 0.809 (95% CI: 0.766–0.848), significantly lower than ReceptorNet ($p < 1 \times 10^{-4}$, DeLong test) (see Fig. 2b).

In addition to ER, the PR− and human epidermal growth factor receptor 2 (HER2)-status may affect tumor growth and thereby may affect the histomorphological structure of the H&E stained tissue. Since HER2 overexpression is a dominant transformation mechanism in tumors[22], discriminative morphological patterns for ERS estimation may be harder to identify in HER2+ samples. Indeed, we find that ReceptorNet performs significantly better ($p < 1 \times 10^{-4}$, F-test) on HER2− samples (AUC = 0.927, 95% CI: 0.912–0.943) as compared to HER2+ samples (AUC = 0.768, 95% CI: 0.719–0.813). Moreover, ReceptorNet performs significantly better ($p < 1 \times 10^{-4}$, F-test) on PR+ samples (AUC = 0.906, 95% CI: 0.869–0.940) as compared to PR− samples (AUC = 0.827, 95% CI: 0.795–0.855), which is reflective of the high correlation between ER and PR statuses (see Fig. 2c). For completeness, we also trained and evaluated ReceptorNet on PR and HER2 labels using the same cross-validation and test split as ERS estimation. ReceptorNet obtains an AUC of 0.810 (95% CI: 0.769–0.846) on PR and an AUC of 0.778 (95% CI: 0.730–0.825) on HER2.

We also find that AUC varies significantly with tumor grade ($p < 1 \times 10^{-3}$, F-test), 0.949 for grade 1 (95% CI: 0.925–0.973), 0.810 for grade 2 (95% CI: 0.716–0.888), and 0.865 for grade

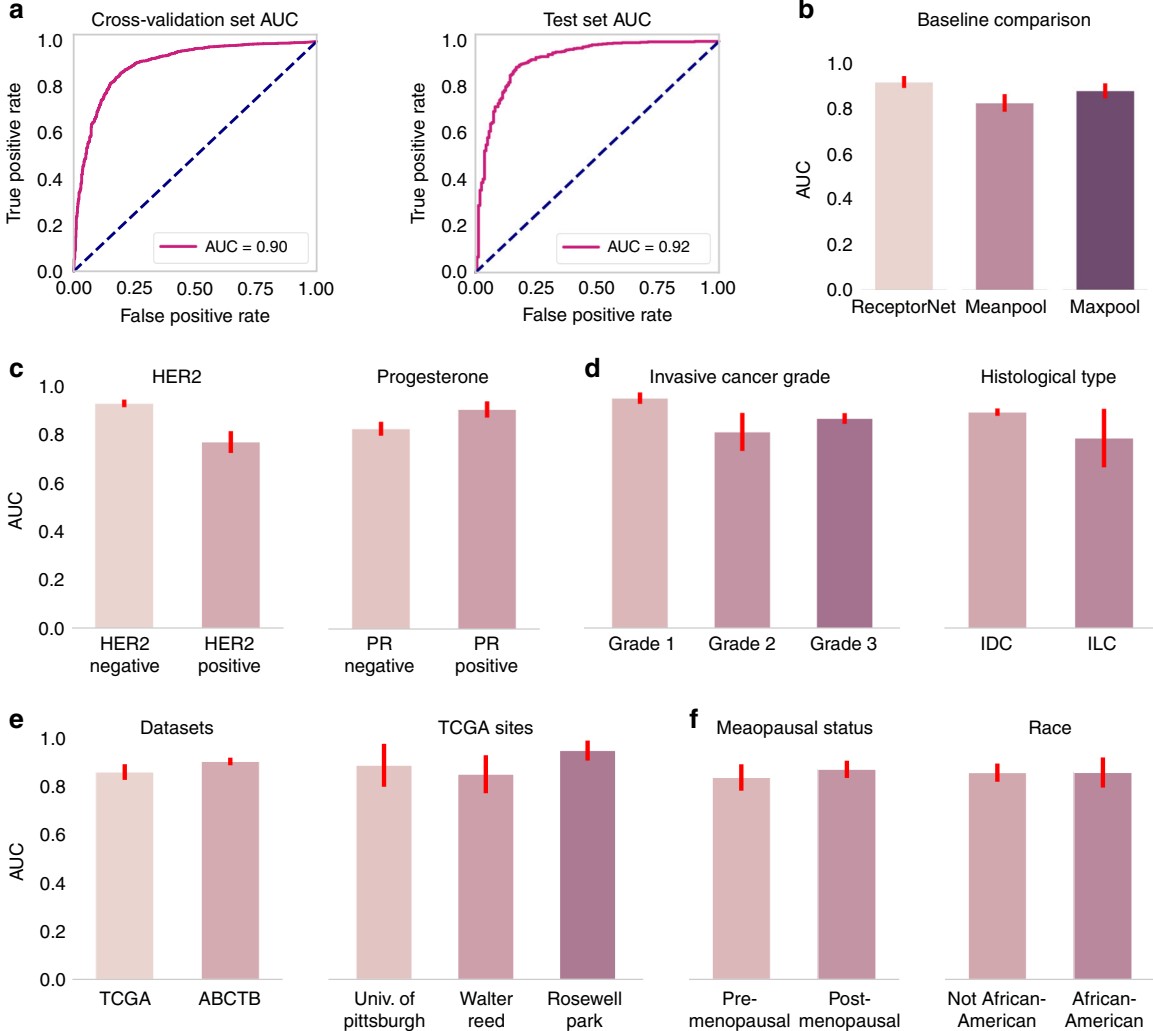

**Fig. 2 ReceptorNet obtains high performance on estrogen receptor status estimation. a** ReceptorNet obtains an AUC of 0.90 on the cross-validation of the train set ($N = 2728$) and an AUC of 0.92 on the test set ($N = 671$). **b** ReceptorNet also beats multiple instance learning baselines on the test set ($N = 671$). On the cross-validation of the train set, ReceptorNet performance is significantly affected by the presence or absence of other hormonal receptors (HER2−: $N = 2205$, HER2+: $N = 407$; PR−: $N = 852$, PR+: $N = 1871$) (**c**), and tumor grade (Grade 1: $N = 297$, Grade 2: $N = 788$, Grade 3: $N = 888$), but not by histological type (IDC: $N = 2073$, ILC: $N = 371$) (**d**). There is no statistically significant difference in ReceptorNet performance across datasets (TCGA: $N = 700$, ABCTB: $N = 2028$), tissue source sites (University of Pittsburgh: $N = 106$, Walter Reed: $N = 79$, Roswell Park: $N = 71$) (**e**) or demographic splits (premenopausal: $N = 250$, postmenopausal: $N = 450$, Not African-American: $N = 583$, African-American: $N = 117$) (**f**). Error bars represent 95% confidence interval for the true AUC calculated by bootstrapping the cross-validation set. Statistical tests for differences in AUC were performed using an upper tail $F$-test. Comparisons between different ERS prediction methods' AUCs on the same data set were performed using the DeLong method.

3 (95% CI: 0.84–0.887). In contrast, we do not find a statistically significant difference in prediction performance based on whether the tumor was ductal or lobular in origin (see Fig. 2d).

There is no statistically significant difference ($p > 0.05$, $F$-test) in ReceptorNet performance across data from Australia, Germany, Poland, and USA. On the cross-validation of the train set, the AUC on TCGA is 0.861 (95% CI: 0.828–0.893); AUC on ABCTB is 0.905 (95% CI: 0.889–0.921). Moreover, ReceptorNet performance across 42 TCGA tissue source sites is similar and is not dependent on the proportion of training samples collected from a given site ($R^2 = 0.16$, $p > 0.1$) (see Fig. 2e).

As additional validation, we trained ReceptorNet only using data from ABCTB and evaluated it on the entire TCGA dataset. ReceptorNet obtains an AUC of 0.850 (95% CI: 0.830–0.868) on TCGA for ERS estimation, a reasonable drop in performance as compared to the combined dataset. TCGA is a harder dataset for prediction as compared to ABCTB, as TCGA has much larger

variation in staining and patient demographics. The ABCTB data is obtained from 6 tissue source sites from one state in Australia, while the TCGA data is obtained from 42 different tissue source sites from the USA, Poland, and Germany. Finally, we removed the data from only the University of Pittsburgh from the training set, which is the largest cohort in TCGA ($N = 134$), and tested ReceptorNet on the data from this site. ReceptorNet obtains an AUC of 0.910 (95% CI: 0.836–0.969) on this cohort, comparable to the AUC obtained on the entire test set (0.920).

We also do not find significant differences in prediction performance based on menopausal status or race ($p > 0.05$, $F$-test) on the TCGA data (demographic information was not available for ABCTB). AUC for postmenopausal women is 0.872 (95% CI: 0.832–0.908). AUC for premenopausal women is 0.838 (95% CI: 0.779–0.893). AUC for African-American patients is 0.859 (95% CI: 0.785–0.921). AUC for the rest of the patients is 0.858 (95% CI: 0.817–0.896) (see Fig. 2f). Trends on test set are similar for

**Table 1 ReceptorNet performs well across different splits of the data based on other patient characteristics and also outperforms baseline methods.**

| Algorithm (data split) | AUC (95% CI) | Algorithm (data split) | AUC (95% CI) |
|---|---|---|---|
| ReceptorNet (cross-validation set) | 0.899 (0.884–0.913) | ReceptorNet (test set) | 0.920 (0.892–0.946) |
| *Baselines* | | | |
| Meanpool (test set) | 0.827 (0.786–0.866) | Maxpool (test set) | 0.880 (0.846–0.912) |
| Individual patch (test set) | 0.760 (0.726–0.794) | Logit on type and grade (test set) | 0.809 (0.766–0.848) |
| *Data splits based on other hormone receptors and grade* | | | |
| ReceptorNet (HER2+, test set) | 0.768 (0.719–0.813) | ReceptorNet (HER2−, test set) | 0.927 (0.912–0.943) |
| ReceptorNet (PR+, test set) | 0.906 (0.869–0.940) | ReceptorNet (PR-, test set) | 0.827 (0.795–0.855) |
| ReceptorNet (Grade 1, test set) | 0.949 (0.925–0.973) | ReceptorNet (Grade 2, test set) | 0.810 (0.716–0.888) |
| ReceptorNet (Grade 3, test set) | 0.865 (0.840, 0.887) | | |
| *Data splits based on data source* | | | |
| ReceptorNet (TCGA, cross-validation set) | 0.861 (0.828–0.893) | ReceptorNet (TCGA, trained on ABCTB alone) | 0.850 (0.830–0.868) |
| ReceptorNet (ABCTB, cross-validation set) | 0.905 (0.889–0.921) | ReceptorNet (University of Pittsburgh, trained on rest) | 0.910 (0.836–0.969) |
| *Data splits based on demographics* | | | |
| ReceptorNet (postmenopausal women, TCGA) | 0.872 (0.832–0.908) | ReceptorNet (premenopausal women, TCGA) | 0.838 (0.779–0.893) |
| ReceptorNet (African-American patients, TCGA) | 0.859 (0.785–0.921) | ReceptorNet (Non-African-American patients, TCGA) | 0.858 (0.817–0.896) |

cohort splits based on hormonal receptors, tumor origin location, demographic variables, and data source (Supplementary Fig. 2). All experimental results are summarized in Table 1.

**Qualitative evaluation**. Next, we evaluate which histomorphological patterns are important to ReceptorNet for ERS estimation. An expert breast cancer pathologist manually reviewed groups of high-attention tiles clustered based on their image features and sorted by their ER+ percentage (see "Methods").

The first group of ER+ discriminative tiles consisted of uniform cells with small nuclei, negligible to modest nuclear pleomorphism with little variation in chromatin pattern, and low mitotic rate, all of which are characteristic features for low grade tumors (Fig. 3a). In contrast, a group of ER− discriminative tiles consisted of nuclei with moderate to substantial nuclear pleomorphism, a relative lack of gland formation, and rapidly growing tumor, which are characteristic of high grade tumors (Fig. 3e). The second group of ER+ discriminative tiles consisted of cells arranged in linear arrays surrounded by stroma, with variation in nuclear size and shape, and no duct formation (Fig. 3b). These are characteristic features for invasive lobular carcinoma (ILC), which validates prospective studies reporting[23–25] ILC to be predominantly ER+. In contrast, none of the tiles with high ER− discriminative ability displayed characteristic patterns for ILC. In the third ER+ discriminative group, tiles were located within ductal/lobular carcinoma in situ lesions composed of small, uniform tumor cells having modest pleomorphism and without intervening stroma (Fig. 3d).

Other groups of ER+ discriminative tiles captured motifs such as invasive tumor cells with intervening reactive stroma (consisting of cancer-associated fibroblasts and myofibroblasts), and a variable number of inflammatory cells. These invasive carcinoma cells had mild to moderate nuclear pleomorphism with interspersed connective tissue, composed predominantly of collagen (Fig. 3c). A group of ER− discriminative tiles contained necrotic debris with reactive lymphoid cells and macrophages removing the debris (Fig. 3f).

In sum, ReceptorNet discovers that histomorphological patterns that identify low grade tumors, ILC, and in situ carcinoma are predictive of ER-positivity. These features have been found to be statistically associated[26–30] with ER+ breast cancers rather than ER−, thus providing validation of some features assessed by the network in determining ERS.

ReceptorNet assigned low attention weights to tiles with fat tissue; tiles with connective tissue with no/few tumor cells; tiles with few tumor cells and reactive stroma trapped in-between fat cells; and tiles with macrophages laden with debris and fat (Fig. 3g). The model automatically learnt to ignore these morphological patterns while making ERS decisions, without any manual pixel-level annotations used for training (Fig. 3h, Supplementary Fig. 3).

We also visualize the learned feature space of the aggregated feature vectors of bags of tiles using t-SNE[31], which shows that ReceptorNet learns to separate WSI based on the degree of ER-positivity (Supplementary Fig. 4).

## Discussion

In conclusion, we demonstrate accurate ER receptor status estimation from H&E stains using a deep neural network trained on a substantial, multi-country dataset of H&E and IHC-labeled image pairs. We test the robustness of our algorithm by varying the presence of other hormonal receptors (PR, human epidermal growth factor), tumor grade, tumor origin location (ductal, lobular), as well as demographic variables (menopause, race), and find that other receptors and tumor grade can significantly influence classifier predictions, while the location of cancer origin and demographic variables considered do not. The ability to determine IHC-derived molecular marker status from H&E stains has the potential to reduce variability in predictions and decrease the cost of pathology workflows. In addition, the time to treatment initiation would be expedited by using a digital workflow, which may affect clinical outcomes[32,33]. In this work, we lay a foundation for future studies to compare the clinical workflow of a pathologist with and without this type of ML. More broadly, our study represents an enhancement of standard physician skill sets and demonstrates ML's potential to improve cancer prognosis and theragnosis by harnessing biological markers currently imperceptible to clinicians.

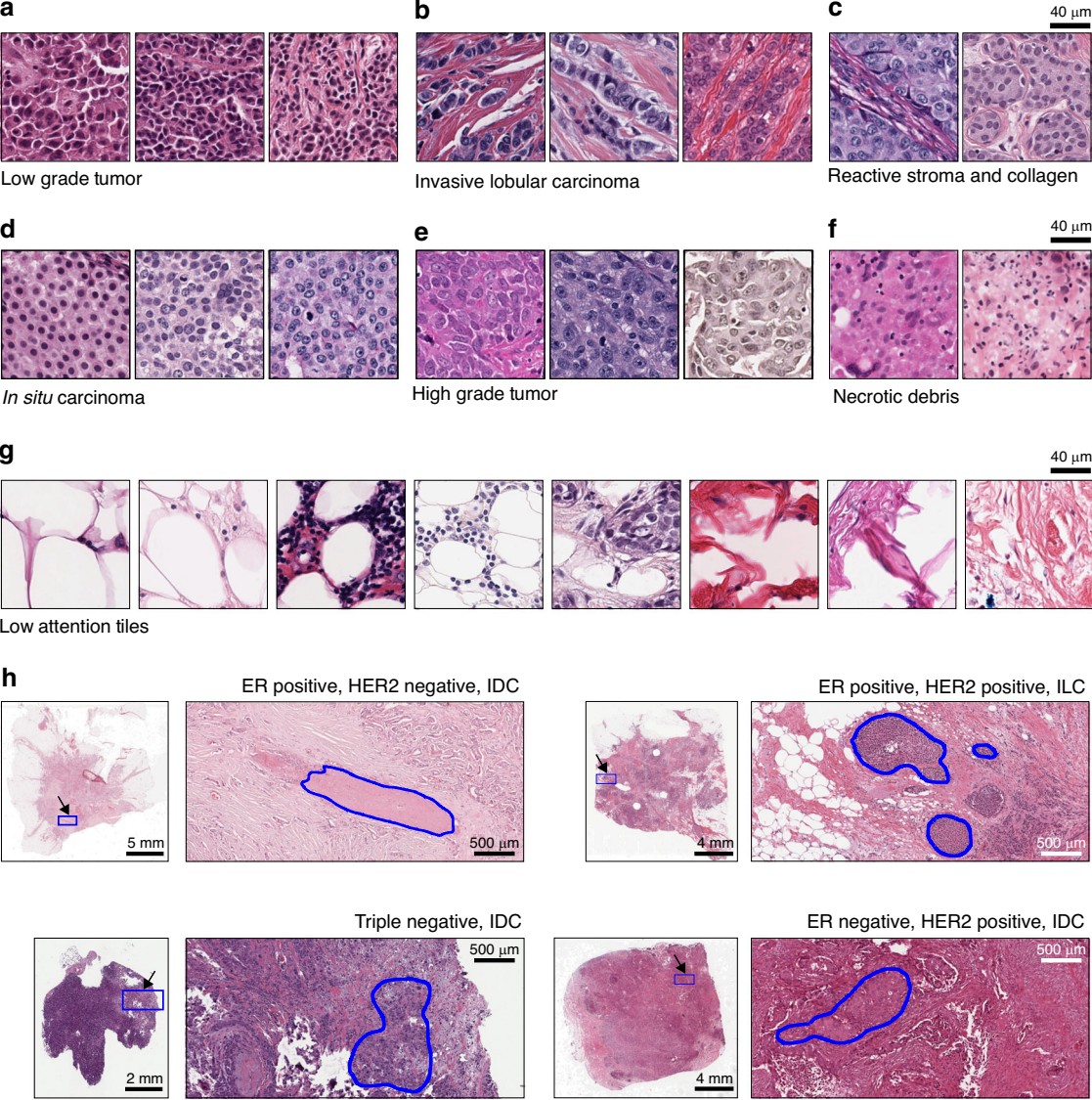

**Fig. 3 ReceptorNet discovers histomorphological patterns important for estrogen receptor status estimation.** ReceptorNet automatically learns that **a** low grade tumors, **b** invasive lobular carcinoma, **c** reactive stroma, and **d** in situ carcinoma are predictive of ER-positivity; and **e** high grade tumor and **f** necrotic debris are predictive of ER-negativity. **g** ReceptorNet also learns to ignore fat tissue; connective tissue with no/few tumor cells; and macrophages laden with debris and fat. **h** Attention weights allow us to identify the regions of whole-slide images important for decision-making. Regions bounded by the blue outline were used for ERS estimation, while other regions were ignored.

## Methods

**Datasets**. We combined hematoxylin and eosin (H&E)-stained whole slide images from two datasets: the Australian Breast Cancer Tissue Bank (ABCTB) dataset, which contains 2535 H&E images from 2535 patients and The Cancer Genomic Atlas (TCGA) dataset, which contains 1014 H&E images from 939 patients. Both datasets report ER, PR, and HER2 status determined by pathologists from IHC. The WSI were scanned at a resolution of 20× or higher.

**Data preparation**. To train ReceptorNet with slide regions that only contain tissue matter, we first performed segmentation using the Otsu's method[34] on the WSI thumbnail and discarded the background regions. We then extracted 256 × 256 pixel image tiles from the foreground of WSI at 20× magnification, where an image was considered to be part of the foreground if 1% of the tile was considered to be tissue matter. The tiles were extracted without any overlap between adjacent tiles. The average number of tiles per slide was 19,944, with the number of tiles per slide varying between 949 and 67,368.

**Model architecture**. The ReceptorNet architecture consists of three interconnected neural networks: a feature extractor module, an attention module, and a decision module. The feature extractor is a ResNet-50[35] without the softmax layer, followed by two fully connected layers with a dropout of 0.5 which convert the 1000-dimensional feature obtained from ResNet-50 to a 512-dimensional feature vector. The ResNet-50

is initialized from ImageNet[36]-pretrained weights and the fully connected layers are randomly initialized using He initialization[35]. During one iteration of training, a bag of $N$ image tiles is fed to the feature extractor, which outputs a $N \times 512$ dimensional feature matrix. This feature matrix is fed to the attention module for aggregation. The first part of the attention module contains a linear layer which reduces each feature vector to 128 dimensions and applies an element-wise hyperbolic tangent ($tanh(.)$) nonlinearity on the output, which scales the features to include values between $-1$ and 1 and facilitates learning of similarities and contrasts between tiles. The output of this linear layer followed by tanh is multiplied by another linear layer and a softmax function which computes an attention weight between 0 and 1 for a particular tile. So, for a bag of $N$ image tiles, we obtain an $N$-dimensional vector of attention weights. We then perform an inner product of this attention vector with the $N \times 512$ dimensional feature matrix to obtain an aggregate 512-dimensional feature vector. The 512-dimensional aggregate feature vector is now fed to a decision layer consisting of a 512-dimensional linear layer followed by a sigmoid function, which outputs a probability between 0 and 1 for a bag of $N$ tiles.

**Training protocol**. We trained ReceptorNet by feeding bags of $N = 50$ tiles drawn randomly from the pre-extracted tiles from each WSI. We performed extensive data augmentation to help the model learn invariances and to deal with variability in staining methods. Specifically, we (i) randomly flipped the tile from left to right with a probability of 0.5, (ii) randomly rotated the tile by {0°, 90°, 180°, 270°} with

equal probability, (iii) performed color jittering, and (iv) performed cutout regularization[18] with length 100. We trained the model to minimize cross-entropy loss using Adam optimizer with a learning rate of $1 \times 10^{-5}$ and weight decay of $5 \times 10^{-5}$ for 500 epochs, employing hard-negative mining[21] during each epoch. Since our dataset contains significant class-imbalance (the number of ER+ samples are 3.7 times the number of ER− samples), we performed balanced sampling to maintain a rough 50–50 proportion of ER+ and ER− samples in each epoch. We found that training the model with a bag containing all the image tiles lead to overfitting. To reduce overfitting, we randomly replaced tiles by an image with all pixel values set to the mean pixel value of the dataset with a probability of 0.75. This mean pixel regularization improved performance substantially. We measured performance using the AUC, PPV, and NPV as metrics for the binary classification task. PPV is defined as (sensitivity × prevalence)/(sensitivity × prevalence + (1 − specificity) × (1 − prevalence)). NPV is defined as (specificity × (1 − prevalence))/((1 − sensitivity) × prevalence + specificity × (1 − prevalence)). The sensitivity and specificity are calculated by binarizing the prediction probability of the network using a specific threshold (in our case, 0.25).

**Comparison to baseline methods**. We compared our method with two widely used MIL methods: Meanpool and Maxpool. In Meanpool, the feature representations of $N$ tiles in a bag are averaged to obtain an aggregate feature representation. In Maxpool, a feature-wise max is obtained for each of the feature dimensions. These methods were trained on the same model architecture as ReceptorNet, except for replacing the Attention module by a Meanpool or Maxpool operation.

**Pathologist review**. We selected a set of highly discriminative tiles for ERS estimation for review by a breast subspecialized pathologist (M.F.P.). We first evaluated our trained model on bags of tiles sampled exhaustively from slides in our test set. From each slide, we saved the 512-dimensional aggregate feature vector obtained from the bag of tiles and performed k-means clustering on the features, with k determined by the elbow method[37]. We computed the ER+ fraction for each cluster using the predicted ER status of the slides in each cluster. For tiles in highly ER+ or ER− clusters (80%+ ER+, or ER−), we performed k-means clustering on features of individual tiles in the top 1% of each slide according to attention weights. We then sorted these tiles by their distance from cluster centers and displayed one tile each from a different slide, displaying up to five tiles from each slide. This exercise ensured selection of highly discriminative tiles from slides with similar aggregate visual representation (and hence ER+ probability) according to our trained algorithm. The pathologist manually reviewed these tiles and recorded observations about the cellular morphology and architecture of the tissue field.

**Hardware and software**. Experiments were performed on USC's high-performance computing cluster, consisting of Nvidia P-100 Pascal graphics processing units (GPUs). Image tiles were extracted using the Python version of the OpenSlide library (v3.4.1). Each model was trained on a single GPU using the PyTorch library (v0.4.1). Performance evaluation, including AUC calculation and CI estimation with bootstrapping, was done using scikit-learn (v0.20.0) library in Python. Statistical tests were performed in R (v3.6.1); the DeLong test was performed using the Daim R package (v1.1.0). All code was developed using open-source tools.

**Statistical methods**. Statistical tests for differences in AUC were performed using an upper tail $F$-test. The numerator variance was computed as the AUC variance across comparison groups. The denominator variance was computed as the pooled variances across comparison groups from 1000 bootstrap AUC values within each group. The numerator degrees of freedom (dof) was taken as the number of groups minus one, and the denominator dof was taken as infinity. The 95% CI calculations were also performed using 1000 bootstrap AUC values within each group. Comparisons between different ERS prediction methods' AUCs on the same data set were performed using the DeLong method[38]. Statistical analysis was done using the following Python libraries: Jupyter (v4.4.0), numpy(v1.15.4), scipy (v1.0.0), and pandas (v0.22.0).

**Dataset curation**. We did not curate the dataset, except removing 75 WSI containing excessive pen markings. The dataset faithfully reflects the quality of WSI encountered in real-world clinical scenarios.

**Data protection**. The study was approved by the Institutional Review Board at the University of Southern California.

**Reporting summary**. Further information on research design is available in the Nature Research Reporting Summary linked to this article.

## Data availability
TCGA dataset is publicly available at the TCGA portal (https://portal.gdc.cancer.gov). The ABCTB dataset is available from the Australian Breast Cancer Tissue Bank subject to ethical and scientific approvals (https://abctb.org.au/abctbNew2/ACCESSPOLICY.pdf).

## Code availability
Our code and experiments can be reproduced by utilizing the details provided in the "Methods" section on data preparation, model architecture, and training protocol, and the following open-source libraries/codebases: Data preparation are based on py-wsi (https://github.com/ysbecca/py-wsi). Major components of our model architecture and training protocol can be reproduced using AttentionDeepMIL (https://github.com/AMLab-Amsterdam/AttentionDeepMIL) and cutout regularization (https://github.com/uoguelph-mlrg/Cutout).

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

## Acknowledgements

Tissues and samples were received from the Australian Breast Cancer Tissue Bank which is generously supported by the National Health and Medical Research Council of Australia, The Cancer Institute NSW, and the National Breast Cancer Foundation. The tissues and samples are made available to researchers on a nonexclusive basis. We thank the ABCTB for their assistance in working with this data. We thank Oracle Corporation for providing Oracle Cloud Infrastructure computing resources used in this study. This research was supported in part by a grant from the Breast Cancer Research Foundation (BCRF-18-002). We thank Shubhang Desai for research assistance, Melvin Gruesbeck for help with illustrations, and Vanita Nemali and Audrey Cook for operational support.

## Author contributions

N.N., A.M., N.S.K., and D.R. designed and performed the experiments. N.N., A.M., A.E., D.R., and D.B.A. analyzed the results. M.F.P. reviewed the whole slide images for qualitative analysis. D.B.A and R.S. supervised and managed the project. All authors contributed to preparation of the paper.

## Competing interests

The authors declare no competing interests.
