## [Peer Review File · Nature Communications]

Reviewers' Comments:

Reviewer #1:

Remarks to the Author:

The paper introduces a new method for predicting the ER status solely from hematoxylin and eosin-stained digital whole slide images. The authors are the first to use whole slide images for ERS prediction with over AUC = 0.92. The concept of biomarker prediction from H&E slides, however, is not new.

For examples, Previous publications already demonstrated the ability of prediction mutations in lung cancer from whole slide H&E images.

[*] "Classification and mutation prediction mutation status from lung cancer"

Other publication have shown ERS prediction from TMA in AUC = 0.88.

[**] "artificial intelligence algorithms to assess hormonal status from tissue microarrays in patients with breast cancer"

Another publication reported ACC = 0.84 for ERS prediction from TMA while the AUC is not reported:

[***] "Image analysis with deep learning to predict breast cancer grade, ER status, histologic subtype, and intrinsic subtype"

The paper focus is on improvement of AUC for ER prediction. Such improvement may be attributed to the dataset rather than the deep learning model. Also, since the authors were exposed to the test set during the development of their model, the increased accuracy may be related to testing various settings of the deep learning model. Thus, if the authors claim novelty of the deep learning method, they should at least compare their MIL+attention method to the most basic one of simply extracting patches and predicting ERS directly from these patches. We were not convinced that the novelty lies in the attention and bag of words model.

If the authors claim their novelty for increased adaptation to clinical settings, they should make a comprehensive comparison to traditional IHC, test their system on an independent cohort that is not part of the training, train their system also on PR, HER2 and KI67, and specifically, discuss how to combine their system in practical settings.

Specific comments:

1) "been determined to be ER-negative (ER-) from an IHC stain, we can assume that their H&E WSI contains almost no discriminative features for ER-positivity. Conversely, if the patient has been determined to be ER-positive (ER+), we can assume that at least some regions of their H&E WSI contain discriminative features for ER-positivity."

The authors assume asymmetry between ER- and ER+ indicative features. More specifically, they assume there are morphological features that indicate ER+, and lack of features that indicate ER-. The authors then justify using MIL based on this axiomatic assumption. The morphological features, however, do not necessarily correspond to IHC receptor staining, and such an assumption needs to be better justified. Moreover, we know this is not entirely true - there are indeed morphological features indicative of ER-. The authors even refer to it in Figure 3 "necrotic debris are predictive of ER-negativity".

2) "On the test set, our method obtains a PPV of .932 and an NPV of .741 at a threshold of .25, comparable to traditional IHC (PPV = .92 and NPV = .683)."

If the authors want to compare their system's accuracy to traditional IHC, they should make a comprehensive comparison:

- The authors should compare also sensitivity and specificity.

- The PPV/NPV of traditional IHC are higher, to the best of our knowledge. Other papers show different results. For example:

[*] "Quality assessment of estrogen receptor and progesterone receptor testing in breast cancer"

using a tissue microarray based approach"

[*] "Comparison of Core Needle Biopsy and Surgical Specimens in Determining Intrinsic Biological Subtypes of Breast Cancer with Immunohistochemistry"

The last table in the paper cited as [4].

- The comparison is between the system's result and the ER annotations labeled by a pathologist (that may contain errors), and not the system's result and the "true" ER status (which is unknown).

- Errors in ER diagnosis may be due to various reasons such as fixation problems, heterogeneity of the tumor, pathologists interpretability, and different protocols between labs. A fair comparison should be on the same dataset and to some "ground truth" annotation that is known to be more accurate than both.

3) If the authors had PR and Her2 annotations, why didn't they train to test their system and predict them?

4) The annotations themselves may contain errors. Were the authors exposed to the IHC images of the patients? If so, did they examine these images to estimate their reliability and the pathologists' interpretability?

5) Figure 4 is nice and may help in general understanding of the machine learning concept. However, beyond that, the value of Figure 4 to the specific idea presented in the paper is unclear.

6) The actual percentage of ER may be important in some clinical scenarios. Is it possible to predict the actual percentage or only the binary status?

7) The benefits of attention learning idea are clear, but this process may also result in loss of information. Patches that are less correlative to the ER status may contribute as well to the overall accuracy. To justify the choice of attention model, the authors should compare their model to a model with no attention - simply learning from all patches. What about using no bag of words and simply assigning the labels at a patch level? Does the attention and MIL model provide significantly high accuracy than the standard one?

8) "Recent studies have shown promising performance for ERS determination from morphological stains, but are based on single center datasets of tissue microarrays (TMAs). Creating TMAs is a manual process that requires pathologists to select regions of interest (ROIs) from the whole specimen"

The authors should note that for other molecular biomarkers there are indeed studies on TCGA with multiple centers and whole slide images. For example: "Classification and mutation prediction mutation status from lung cancer"

Also, predicting ERS from WSI may actually be an easier task than predicting from TMA, because TMA contains much less tissue.

9) Since the TCGA contains 40 cohorts, the authors should remove one of the cohorts from training and test their system on it, to show the results on an independent test cohort that was not part of training.

Reviewer #2:

Remarks to the Author:

The authors present a method to predict ER status from H&E stained slides of breast cancer using a multiple instance learning (MIL) based approach, specifically an adapted version of AttentionNet. The method was validated with a relatively large set of H&E slides that are publicly available (most

by request). The performance the authors report is very good and I think the analysis with the pathologist and the interpretable features is a nice addition. The final presentation of the results in which the authors interrogate sources of variation is interesting as well.

However, I think the paper has one major, quite fundamental flaw, which can be addressed by the authors. In several similar studies (which the authors could have done a better job citing, e.g. Veta et al. in *Medical Image Analysis* (2019), Tellez et al. in *TPAMI* (2019), Schaumberg et al. bioRxiv, <https://www.biorxiv.org/content/10.1101/064279v9>, among others), where for example genetic mutations are predicted from H&E images, it is impossible to verify if the found features or relevant areas indeed correspond to mutation. However, in this study it is actually possible to just that. If the authors would collect even a small set of corresponding H&E and ER IHC slides the results from AttentionNet and the morphological features could directly be coupled to areas of ER positivity. This would show conclusively if the network is indeed picking up morphological patterns which are specific to ER positivity, or feature which correlate with other aspects such as grade or type which also correlate with ER positivity.

An additional experiment the authors could also conduct is to train with just the ABCTB data and validate on the TCGA data, this also gives some indication as to how well the features generalize. I know this is challenging due to the staining variation, but this can at least partially be mitigated with data augmentation or maybe one or two of the myriad of normalization algorithms which are out there.

Some smaller comments pertain to the reference standard used for training and validation and the comparison to the traditional IHC. The paper the authors cite (reference 3) actually goes into great length explaining that most of the variation in ER scoring is due to regional differences and that centralized ER scoring is actually pretty consistent. How are the datasets the authors use scored, centrally or regionally? If regionally, the exact issue the authors mention (variability in ER positivity) is also present in both their training and test data. This further calls for additional external validation of their results. Last, the authors mention ..comparable to traditional IHC (PPV = .92 and NPV = .683).. but is unclear where exactly these numbers come from. The reference shows many, many results for IHC with varying reference standards and procedures, so cherry-picking a single operating point seems inappropriate.

Deep learning-enabled breast cancer hormonal receptor status determination from base-level H&E stains

Manuscript Number: NCOMMS-20-15587-T

Point-by-point Responses

Reviewer #1 (Remarks to the Author):

The paper introduces a new method for predicting the ER status solely from hematoxylin and eosin-stained digital whole slide images. The authors are the first to use whole slide images for ERS prediction with over AUC = 0.92. The concept of biomarker prediction from H&E slides, however, is not new.

For examples, Previous publications already demonstrated the ability of prediction mutations in lung cancer from whole slide H&E images.

[*] "Classification and mutation prediction mutation status from lung cancer"

Other publication have shown ERS prediction from TMA in AUC = 0.88.

[**] "artificial intelligence algorithms to assess hormonal status from tissue microarrays in patients with breast cancer"

Another publication reported ACC = 0.84 for ERS prediction from TMA while the AUC is not reported:

[***] "Image analysis with deep learning to predict breast cancer grade, ER status, histologic subtype, and intrinsic subtype"

A: The paper focus is on improvement of AUC for ER prediction. Such improvement may be attributed to the dataset rather than the deep learning model.

-- Our deep learning model statistically significantly improves performance over conventional MIL approaches; on the test set, the AUC for AttentionNet is .093 higher than Meanpool and .04 higher than Maxpool com. So the improvement in performance is not attributable to the dataset alone. Moreover, the AUC remains high on smaller splits of the dataset (e.g., ABCTB or TCGA alone). Finally, reporting performance at larger sample sizes is of importance in medical AI research, since it improves our understanding of how increasing training set size improves performance on the task of interest.

B. Also, since the authors were exposed to the test set during the development of their model, the increased accuracy may be related to testing various settings of the deep learning model.

-- We did not evaluate our model on the test set at any point during the model design and hyperparameter selection process (as is standard practice in machine learning research). Only

the final model selected using the cross-validation set was evaluated on the training set. Therefore, we do not think that the increased accuracy is related to testing various settings of the deep learning model.

C. Thus, if the authors claim novelty of the deep learning method, they should at least compare their MIL+attention method to the most basic one of simply extracting patches and predicting ERS directly from these patches. We were not convinced that the novelty lies in the attention and bag of words model.

Thanks for the excellent suggestion. We trained a basic model for predicting ERS directly from individual patches with the same basic architecture after removing the MIL+attention module. This model obtains an AUC of 0.760 on the test set, much lower than the AUC of 0.920 obtained by AttentionNet. We have updated the wording in our paper to highlight this point.

Further, please note that AttentionNet predicts ERS by aggregating morphological features indicative of grade, histological type, and other predictors from many patches at once. It is unlikely that a single patch would contain the same predictive signal. The Meanpool MIL model reported in the paper determines ERS from multiple patches at once by simply averaging their feature vectors and obtains an AUC of 0.827, significantly lower than AttentionNet. These two experiments indicate that the MIL+ attention method is indeed responsible for the improvement in ERS estimation performance.

D. If the authors claim their novelty for increased adaptation to clinical settings, they should make a comprehensive comparison to traditional IHC, test their system on an independent cohort that is not part of the training, train their system also on PR, HER2 and KI67, and specifically, discuss how to combine their system in practical settings.

The idea of taking this one step further, into a clinical setting and observing its impact on real workflows, is an excellent suggestion, thank you. We will incorporate this into subsequent studies. Following your suggestion, we trained our machine learning system on PR and HER2 (KI67 observations were not available, but we agree this is a key biomarker to consider in future work) and evaluated it on the same test set for ER (which was completely separate from our training set). Our algorithm obtained an AUC of 0.810 on PR and an AUC of 0.778 on HER2. We have included these results in the paper.

Specific comments:

1) "been determined to be ER-negative (ER-) from an IHC stain, we can assume that their H&E WSI contains almost no discriminative features for ER-positivity. Conversely, if the patient has been determined to be ER-positive (ER+), we can assume that at least some regions of their H&E WSI contain discriminative features for ER-positivity."

The authors assume asymmetry between ER- and ER+ indicative features. More specifically, they assume there are morphological features that indicate ER+, and lack of features that indicate ER-. The authors then justify using MIL based on this axiomatic assumption. The morphological features, however, do not necessarily correspond to IHC receptor staining, and such an assumption needs to be better justified. Moreover, we know this is not entirely true - there are indeed morphological features indicative of ER-. The authors even refer to it in Figure 3 "necrotic debris are predictive of ER-negativity".

-- Thank you for the suggestions. We emphasize that our key MIL assumption is that a bag of patches obtained from a slide marked as ER-negative contain no discriminative features for ER-positivity and at least some patches of an ER-positive slide contain discriminative features for ER-positivity. We further clarify that the assumed correspondence is between morphological features visible in H&E, and the IHC-obtained WSI *label*. We do not assume correspondence between visible features of H&E and IHC.

2) "On the test set, our method obtains a PPV of .932 and an NPV of .741 at a threshold of .25, comparable to traditional IHC (PPV = .92 and NPV = .683)."

If the authors want to compare their system's accuracy to traditional IHC, they should make a comprehensive comparison:

- The authors should also compare sensitivity and specificity.

-- We appreciate the suggestion. We have updated this sentence in the paper to clarify that the IHC comparison is based on a separate study (the International Breast Cancer Study Group study reported by Hammond et al. 2010), and not our own.

- The PPV/NPV of traditional IHC are higher, to the best of our knowledge. Other papers show different results. For example:

[*] "Quality assessment of estrogen receptor and progesterone receptor testing in breast cancer using a tissue microarray based approach":

[*] "Comparison of Core Needle Biopsy and Surgical Specimens in Determining Intrinsic Biological Subtypes of Breast Cancer with Immunohistochemistry": (N = 1,371, from Samsung

The last table in the paper cited as [4].

- The comparison is between the system's result and the ER annotations labeled by a pathologist (that may contain errors), and not the system's result and the "true" ER status (which is unknown).

- Errors in ER diagnosis may be due to various reasons such as fixation problems, heterogeneity of the tumor, pathologists interpretability, and different protocols between labs. A

fair comparison should be on the same dataset and to some "ground truth" annotation that is known to be more accurate than both.

-- The authors agree, and have looked into this line of work. The challenge faced is a lack of access to the original tissue material that would be required to perform an exact comparison between pathologists and our approach. Validation against clinical outcomes remains an important future direction.

3) If the authors had PR and Her2 annotations, why didn't they train to test their system and predict them?

-- Our algorithm obtained an AUC of 0.810 on PR and an AUC of 0.776 on HER2.

4) The annotations themselves may contain errors. Were the authors exposed to the IHC images of the patients? If so, did they examine these images to estimate their reliability and the pathologists' interpretability?

-- Unfortunately we did not have access to the IHC images of the patients.

5) Figure 4 is nice and may help in general understanding of the machine learning concept. However, beyond that, the value of Figure 4 to the specific idea presented in the paper is unclear.

-- Thank you for the feedback. We have placed this figure in the supplementary material.

6) The actual percentage of ER may be important in some clinical scenarios. Is it possible to predict the actual percentage or only the binary status?

-- Unfortunately, the information on percentage of ER is not available for a large majority of our dataset. But we agree that this would be an excellent direction for future research.

7) The benefits of attention learning idea are clear, but this process may also result in loss of information. Patches that are less correlative to the ER status may contribute as well to the overall accuracy. To justify the choice of attention model, the authors should compare their model to a model with no attention - simply learning from all patches. What about using no bag of words and simply assigning the labels at a patch level? Does the attention and MIL model provide significantly high accuracy than the standard one?

-- Excellent observation. AttentionNet significantly outperforms models without attention. Please see response to question D, above.

8) "Recent studies have shown promising performance for ERS determination from morphological stains, but are based on single center datasets of tissue microarrays (TMAs). Creating TMAs is a manual process that requires pathologists to select regions of interest (ROIs) from the whole specimen"

The authors should note that for other molecular biomarkers there are indeed studies on TCGA with multiple centers and whole slide images. For example: ""Classification and mutation prediction mutation status from lung cancer"

-- Thank you for the reference. We have now explicitly mentioned this work in the paper.

Also, predicting ERS from WSI may actually be an easier task than predicting from TMA, because TMA contains much less tissue.

-- This is a solid point, and worth considering (and comparing) in future studies. Our goal here was a bit different---since clinical workflows involve a pathologist determining ERS from WSI, we aimed to develop a machine learning algorithm that could aid the physician.

9) Since the TCGA contains 40 cohorts, the authors should remove one of the cohorts from training and test their system on it, to show the results on an independent test cohort that was not part of training.

-- Thank you for this suggestion. We removed the data from the University of Pittsburgh from the training set, which is the largest cohort in TCGA (N = 134), and tested our system on it. AttentionNet obtained an AUC of 0.910 on this cohort, comparable to the AUC obtained on the entire test set (0.920). We have included this result in the paper.

Reviewer #2 (Remarks to the Author):

The authors present a method to predict ER status from H&E stained slides of breast cancer using a multiple instance learning (MIL) based approach, specifically an adapted version of AttentionNet. The method was validated with a relatively large set of H&E slides that are publicly available (most by request). The performance the authors report is very good and I think the analysis with the pathologist and the interpretable features is a nice addition. The final presentation of the results in which the authors interrogate sources of variation is interesting as well.

1. However, I think the paper has one major, quite fundamental flaw, which can be addressed by the authors. In several similar studies (which the authors could have done a better job citing, e.g. Veta et al. in Medical Image Analysis (2019), Tellez et al. in TPAMI (2019), Schaumberg et al. bioRxiv, <https://www.biorxiv.org/content/10.1101/064279v9>, among others), where for example genetic mutations are predicted from H&E images, it is impossible to verify if the found features or relevant areas indeed correspond to mutation. However, in this study it is actually

possible to just that. If the authors would collect even a small set of corresponding H&E and ER IHC slides the results from AttentionNet and the morphological features could directly be coupled to areas of ER positivity. This would show conclusively if the network is indeed picking up morphological patterns which are specific to ER positivity, or feature which correlate with other aspects such as grade or type which also correlate with ER positivity.

-- Thank you for this excellent suggestion. Unfortunately we were unable to access corresponding H&E and IHC slides to perform the suggested analysis. We are looking for collaborators to work on this very important question.

2. An additional experiment the authors could also conduct is to train with just the ABCTB data and validate on the TCGA data, this also gives some indication as to how well the features generalize. I know this is challenging due to the staining variation, but this can at least partially be mitigated with data augmentation or maybe one or two of the myriad of normalization algorithms which are out there.

-- Thank you for this suggestion. We trained with the ABCTB data and validated on the TCGA data using extensive data augmentation to deal with staining variation. Specifically, we performed color jittering, which we have found to provide better results as compared to stain normalization---a finding also reported by e.g., Nagpal et al., 2019. AttentionNet obtained an AUC of 0.850 on TCGA for ERS estimation, a reasonable drop in performance as compared to the combined dataset. TCGA is a harder dataset for prediction as compared to ABCTB, as TCGA has a much larger variation in staining and patient demographics. The ABCTB data is obtained from 6 tissue source sites from one state in Australia, while the TCGA data is obtained from 42 different tissue source sites from the USA, Poland, and Germany. We have included these results in the paper.

3. Some smaller comments pertain to the reference standard used for training and validation and the comparison to the traditional IHC. The paper the authors cite (reference 3) actually goes into great length explaining that most of the variation in ER scoring is due to regional differences and that centralized ER scoring is actually pretty consistent. How are the datasets the authors use scored, centrally or regionally? If regionally, the exact issue the authors mention (variability in ER positivity) is also present in both their training and test data. This further calls for additional external validation of their results.

-- The datasets have been scored regionally. Unfortunately we do not have access to the original IHC images to create a centrally-scored dataset. We agree that external validation with a gold standard dataset is an important future direction.

4. Last, the authors mention ..comparable to traditional IHC (PPV = .92 and NPV = .683).. but is unclear where exactly these numbers come from. The reference shows many, many results for

IHC with varying reference standards and procedures, so cherry-picking a single operating point seems inappropriate.

-- We follow Shamai et al.⁶ and report the PPV and NPV for IHC based on concordance between primary institution and central testing for ER in IBCSG reported in Table 3 of Hammond et al (2010). For our method, the PPV varies between 0.966-0.932 and the NPV varies between 0.507-0.741 for operating points between .75 and .25. We would like to stress that our goal is not to directly compare our accuracy with IHC, since such comparison is indeed challenging to make in the absence of a gold standard dataset. The comparison made here is purely for illustrative purposes. We have now clarified this in the paper.

Reviewers' Comments:

Reviewer #1:

Remarks to the Author:

Thank you for your answers and further experiments following my comments.

1)

You added several more results.

I would suggest summarizing all experimental setups and their results in one table, since I find it confusing for the reader to follow and compare the results in the text. Please add to the table the concordance rates of the methods you compared to.

2)

"On the test set, our method obtains a PPV of .932 and an NPV of .741 at a threshold of .25."

As I understand, the threshold is used as a cut-off for binarizing the system's score, but this is very unclear. Please explain the process of obtaining the PPV/NPV clearly in the text (or in Methods).

"We note that the results from the International Breast Cancer Study Group (3,4) dataset from a separate study, translate to a PPV of .92 and an NPV of .683 at the threshold of .25"

I do not understand this sentence. How were these numbers obtained, and what is the threshold 0.25 in this case?

3)

Thank you for assessing and showing the results of the patch-based deep learning system that reached 0.76 AUC.

I think it would be beneficial to add the results also AFTER averaging patch scores - average at least 500-1000 patch scores per slide (unlike meanPool which as I understand averages 50 patches?). This is what most works are doing and I think it should be included as a baseline for comparison to your attention learning method (and was actually what I meant).

Reviewer #2:

Remarks to the Author:

I think the authors did a pretty good job responding to my critiques. It is unfortunate that no IHC images are available, as this weakens the section on the interpretability significantly. However, the rest of the paper is strong enough to warrant publication in my view. The results also, mostly, hold up when switching the different centers or excluding a center from TCGA, which shows the generalizability of the algorithm. The only thing I would like the authors to address is that in response to the comment from the other reviewer on a patch-based baseline, the authors do not explain how the aggregation is done in this case. I'm guessing some form of mean or max-pooling across the tiles is used for the features. I think both options should be tested and the best reported.

Deep learning-enabled breast cancer hormonal receptor status determination from base-level H&E stains

Manuscript Number: NCOMMS-20-15587-B

Second Revision: Point-by-point Responses (in brown)

Reviewer #1 (Remarks to the Author):

Thank you for your answers and further experiments following my comments.

1) You added several more results.

I would suggest summarizing all experimental setups and their results in one table, since I find it confusing for the reader to follow and compare the results in the text. Please add to the table the concordance rates of the methods you compared to.

-- Thank you for this suggestion. We have now added a table (Table 1) summarizing all experimental setups and other methods that we have compared with.

2) "On the test set, our method obtains a PPV of .932 and an NPV of .741 at a threshold of .25." As I understand, the threshold is used as a cut-off for binarizing the system's score, but this is very unclear. Please explain the process of obtaining the PPV/NPV clearly in the text (or in Methods).

-- Thank you for this suggestion. We have now included a description of the process of obtaining the PPV and NPV in Methods.

"We note that the results from the International Breast Cancer Study Group (3,4) dataset from a separate study, translate to a PPV of .92 and an NPV of .683 at the threshold of .25" I do not understand this sentence. How were these numbers obtained, and what is the threshold 0.25 in this case?

-- We follow Shamaï et al.⁶ and report the PPV and NPV for IHC based on concordance between primary institution and central testing for ER in IBCSG reported in Table 3 of Hammond et al (2010). The words "at the threshold of .25" were included here by mistake and have been deleted. Thank you for spotting this.

3) Thank you for assessing and showing the results of the patch-based deep learning system that reached 0.76 AUC.

I think it would be beneficial to add the results also AFTER averaging patch scores - average at least 500-1000 patch scores per slide (unlike meanPool which as I understand averages 50 patches?). This is what most works are doing and I think it should be included as a baseline for comparison to your attention learning method (and was actually what I meant).

-- Thank you for your suggestion. The results reported for the patch-based deep learning system during the last revision were averaged over 50 patches during inference. We have now computed the performance when using the average of 1000 patch scores, and the score only improves modestly to an AUC of 0.781, as compared to an AUC of 0.760 for 50 patches. We have observed this phenomenon in Meanpool as well, in that there is a diminishing return in performance improvement with increasing number of patches.

Reviewer #2 (Remarks to the Author):

1. I think the authors did a pretty good job responding to my critiques. It is unfortunate that no IHC images are available, as this weakens the section on the interpretability significantly. However, the rest of the paper is strong enough to warrant publication in my view. The results also, mostly, hold up when switching the different centers or excluding a center from TCGA, which shows the generalizability of the algorithm.

-- Thank you for the feedback. We concur that improving interpretability with IHC images remains an important future direction.

The only thing I would like the authors to address is that in response to the comment from the other reviewer on a patch-based baseline, the authors do not explain how the aggregation is done in this case. I'm guessing some form of mean or max-pooling across the tiles is used for the features. I think both options should be tested and the best reported.

-- For the patch-based baseline, we train the network on individual patches, using slide-level labels for ER-positivity. At inference time, we average the probability scores for each patch, as predicted by the network. We do not aggregate at the feature-level, as is the case for meanpool and maxpool baselines reported in the paper.